# The Structural Diversity and Biological Activity of Steroid Oximes

**DOI:** 10.3390/molecules28041690

**Published:** 2023-02-10

**Authors:** Ana R. Gomes, Ana S. Pires, Fernanda M. F. Roleira, Elisiário J. Tavares-da-Silva

**Affiliations:** 1Univ Coimbra, CIEPQPF, Faculty of Pharmacy, Laboratory of Pharmaceutical Chemistry, Azinhaga de Santa Comba, Pólo III - Pólo das Ciências da Saúde, 3000-548 Coimbra, Portugal; 2Univ Coimbra, Coimbra Institute for Clinical and Biomedical Research (iCBR) area of Environment Genetics and Oncobiology (CIMAGO), Institute of Biophysics, Faculty of Medicine, Azinhaga de Santa Comba, Pólo III - Pólo das Ciências da Saúde, 3000-548 Coimbra, Portugal; 3Clinical Academic Center of Coimbra (CACC), Praceta Professor Mota Pinto, 3004-561 Coimbra, Portugal; 4Univ Coimbra, Center for Innovative Biomedicine and Biotechnology (CIBB), Rua Larga, 3004-504 Coimbra, Portugal

**Keywords:** steroids, oximes, chemistry, antitumor, anti-inflammatory, antibacterial, antifungal, antiviral

## Abstract

Steroids and their derivatives have been the subject of extensive research among investigators due to their wide range of pharmacological properties, in which steroidal oximes are included. Oximes are a chemical group with the general formula R_1_R_2_C=N−OH and they exist as colorless crystals and are poorly soluble in water. Oximes can be easily obtained through the condensation of aldehydes or ketones with various amine derivatives, making them a very interesting chemical group in medicinal chemistry for the design of drugs as potential treatments for several diseases. In this review, we will focus on the different biological activities displayed by steroidal oximes such as anticancer, anti-inflammatory, antibacterial, antifungal and antiviral, among others, as well as their respective mechanisms of action. An overview of the chemistry of oximes will also be reported, and several steroidal oximes that are in clinical trials or already used as drugs are described. An extensive literature search was performed on three main databases—PubMed, Web of Science, and Google Scholar.

## 1. Introduction

Steroids belong to a class of natural or synthetic organic compounds, whose basic molecular structure is typically composed of 17 carbon atoms, bonded in four “fused” rings: three six-member cyclohexane rings (rings A, B and C) and one five-member cyclopentane ring (the D ring). (Figure 1). They play a crucial role in the human body, being responsible for the regulation of several biological processes. This fact, together with their interesting biochemical properties, such as the ability to penetrate cell membranes and bind to the nuclear and membrane receptors, makes them extremely attractive in the design of new potential drugs for the treatment of several diseases [1,2]. In fact, since their discovery in 1935, steroids have been widely used in the treatment of several conditions in the most variable areas of medicine, for example, for the treatment of autoimmune and inflammatory diseases and for the treatment of cancer [3,4]. Given the privileged scaffold of steroids and their suitability for structural modifications, steroidal derivatives have been arousing interest among medicinal chemists in the hunt for novel drug candidates. Slight alterations in the basic ring structure of steroids can elicit an enormous change in biological activity, giving rise to steroidal derivatives with a wide range of therapeutic activities [5,6].

Oximes are one of the most popular and extensively hailed nitrogen-containing biological compounds, presenting several biological and pharmacological applications [7]. They have achieved popularity due to their application as antidotes against nerve agents, which is attained by their capacity to reactivate acetylcholinesterase (AChE) [8]. Since that, these hydroxy-imine derivatives have also been associated with several other biological activities such as antibacterial, antifungal, anti-inflammatory, antioxidant, and finally, anticancer as described [7,9].

Employing the hydrophobic steroid core with a hydroxyimino group constitutes an advantage since this chemical group can increase the molecules’ ability to interact with cell membranes, paving the way for enhanced biological activity [10]. For these reasons, in the last 20 years, a reasonable number of new steroidal oximes has been designed and synthesized and then evaluated for their biological activity. This review will focus mainly on the most active steroidal oximes developed as antitumor and antimicrobial agents. Additionally, a few examples of steroidal oximes with anti-inflammatory activity are also considered.

## 2. Chemistry of Steroidal Oximes

Oximes (R_1_R_2_C=N−OH) (Figure 2) are a nitrogen-rich group of compounds, which are produced in nature in the plant and animal kingdom. In plants, oximes and their derivatives play a fundamental role in the metabolism of plant growth and development and in a variety of biosynthetic pathways [9,11]. In animals, oximes are most commonly known for their participation in the olfactory communication between animals [12]. Oximes exist as colorless crystals, are poorly soluble in water, and are easily accessible in laboratories and in industry, which makes them very appealing [13,14]. Additionally, they are extensively used not only as protectors of carbonyl groups but also as intermediates in the Beckmann rearrangement to synthesize several lactam derivatives [9,15,16]. Furthermore, oximes have the particularity of being easily transformed into different chemical groups such as amines, nitro, and other heterocyclic compounds [16,17].

There are several ways to produce oxime derivatives and some reviews have been published regarding the chemistry of oximes [13,16,18]. The most classical method of oxime synthesis, which is the most used in the synthesis of steroidal oximes, involves the reaction of a carbonyl compound, a ketone or an aldehyde with hydroxylamine (NH_2_OH) or a hydroxylammonium salt in the presence of a base (Figure 3). This type of reaction with aldehydes and non-symmetrical ketones can originate the two *E* or *Z* isomeric oxime forms, which can exist both as single compounds and or in mixture. Such chemical aspects can have a great impact on biological activity [7,12].

Apart from the most commonly used synthetic strategy, there are other methods to prepare oximes involving non-carbonyl compounds, which consist of the reduction of nitroalkenes to create aldoximes and ketoximes. The reduction of α,β-unsaturated nitroalkenes gives rise to different oxime derivatives, depending if the nitro group is terminal or internal. More specifically, when the nitro group is terminal, aldoximes are produced, in mildly acidic conditions, in good yields. On the contrary, if the nitro group is internal, under basic conditions ketoximes are formed also in good yields [9,16]. Other variants of oxime synthesis include oxime ethers, esters, and amidoximes, all of which are of great biological and pharmacological importance. Oximes can act both as weak acids and weak bases. For this reason, the oxime anions can behave as ambident nucleophiles, which means that they can attack through two different sites, allowing them to be widely used for the synthesis of the above-mentioned class of compounds (ethers, nitrones, etc.) [13].

Another aspect of this chemical group is that it can behave both as hydrogen-bond donor (via OH group) and as hydrogen-bond acceptor (via nitrogen and oxygen atoms), which together with the high polarity of the oxime moiety can have a tremendous impact on the interaction with the receptor binding sites, enhancing biological activity, when compared to the carbonyl group [7,9]. This premise will be the focus of our review, in which we discuss several synthesized steroidal oximes with enhanced biological activity when compared with the parent carbonyl compounds.

## 3. Steroidal Oximes as Antitumor Agents

Steroidal compounds have been associated with antitumor activity for many years. Several reports focusing on their interesting properties as anticancer agents have been published [19,20,21,22,23,24,25,26,27,28,29,30,31,32,33]. The oxyimino group of oxime compounds is also a structural feature that confers very interesting biological properties among them antitumor activity [7,34]. For this reason, cytotoxic steroidal oximes have been extensively studied throughout the years [14,34]. In this review, we focused on the most active steroidal oximes, described in the literature, against several types of cancer. They are divided according to their steroidal motif: androstane, estrane, pregnane, cholestane, diosgenin, and bile acid derivatives.

### 3.1. Androstane Derivatives

5α-Reductase inhibitors have been widely studied for the treatment of diseases that are exacerbated by 5α-dihydrotestosterone (DHT). Finasteride and dutasteride are two 5α-reductase inhibitors approved by the FDA with several therapeutic applications such as for the treatment of benign prostatic hyperplasia and prostate cancer. Bearing this in mind, Dhingra et al. developed a series of 17-oxyimino -5-androsten-3β-yl esters and evaluated their cytotoxic activity against prostate cancer cells [35]. Compound **1a** (Figure 4) was the most active in DU145 prostate cancer cells with a percentage of growth inhibition of almost 91% at 5 µg/mL and an IC_50_ value of 3.8 µM (Table 1), being more active than finasteride (78.51% of growth inhibition and IC_50_ = 3.9 µM). Moreover, the authors also evaluated the compounds’ acute toxicity using mouse macrophages, which indirectly allowed them to test for the compounds’ selectivity towards cancer cells. The toxicity index value (LC_50_) presented by compound **1a** was very high (LC_50_ = 89.4 µM), proving that **1a** was non-toxic to mouse macrophages.

The introduction of a heteroatom or the substitution of one or more carbon atoms in the steroid scaffold by a heteroatom can have a great impact on the compound’s biological activity. Aza-homosteroids are a class of compounds with unusual structures, which have been associated with a wide range of biological activities such as antiparasitic, antifungal, and anticancer [36,37,38]. Following this line, Huang et al. synthesized a series of 17a-aza-D-homoandrostan-17-one derivatives, namely oximes **1b** and **1c** (Figure 4) [39]. Further cytotoxicity analysis by the 3-(4,5-dimethylthiazol-2-yl)-2,5-diphenyltetrazolium bromide (MTT) method revealed that both compounds were active against HeLa and SMMC7404 cells, being that compound **1b** stood out in HeLa cells with an IC_50_ of 15.1 µM (Table 1). Moreover, these oxime derivatives were in general more active than the compounds without the oxyimino group in their structure, which proves that this chemical group is important in conferring cytotoxicity.

Gomes et al. [34] designed and synthesized two novel steroidal oxime derivatives and evaluated them and two other previously synthesized oximes [40,41] in several cancer cell lines to assess their antiproliferative profile. Initial screening in WiDr, PC3, HepG2, and H1299 cancer cell lines revealed that oximes **1d** and **1e** (Figure 4) were able to decrease all cancer cells proliferation, being especially active against PC3 (IC_50_ = 13.8 µM for **1d** and 14.5 µM for **1e**) and WiDr (IC_50_ = 9.1 for **1d** and 16.1 µM for **1e**) cells (Table 1). Moreover, both oximes were even more potent than some of the chemotherapeutic drugs currently in clinical use for these types of cancer. Both oximes were able to induce cell cycle arrest at different phases, accompanied by cell death by apoptosis/necroptosis and oxidative stress (detected by an increase in ROS production) in both cell lines. Selectivity against cancer cells was also assessed by testing the compounds in normal human colon cells. Results demonstrated that both compounds are selective toward colon cancer cells [34].

Several novel oxime derivatives, such as (17*E*)-(pyridin-2-yl)methylidene 3-oximes **1f**–**1h** were designed and synthesized (Figure 4) [42]. After the synthesis, the antiproliferative activity of both compounds was assessed in a series of several human cancer cell lines (MCF7, MDA-MB-231, PC3, HeLa, HT29, A549) and a normal human cell line, MRC5. A549, HT29, and MDA-MB-231 cancer cells were the most sensitive cell lines to both oximes being that A549 was the one with the best IC_50_ values for all compounds, **1f** (IC_50_ = 1.5 µM) **1g** (IC_50_ = 1.8 µM) and **1h** (IC_50_ = 2.0 µM) (Table 1). Apoptosis induction analysis showed that these oximes induced apoptosis in A549 cells, while at the same time being non-toxic to normal lung fibroblasts MRC5. These results were very encouraging since lung cancer remains one of the most difficult cancers to treat.

Some steroidal compounds with a hydroxyimino group at position C-7 conjugated with an α,β-double bond in position C-5 were designed and synthesized [43]. After the synthesis, their antitumor activity was evaluated against several types of cancer such as cervical, gastric, epidermoid, and breast cancer. Results demonstrated that compounds **1i** and **1j** (Figure 4) were both able to decrease KB, HeLa, MKN-28, and MCF7 cancer cell proliferation (Table 1) and were more active than the corresponding parent ketones. Moreover, compound **1i** was especially active against MCF7 cells presenting an IC_50_ value of 10.2 µM.

Dubey et al. designed and synthesized novel dioximes of 16-benzylidene substituted derivatives [44]. The antitumor activity of these dioximes was then evaluated in terms of the percentage of growth inhibition of NCI-H460, MCF7, and SF268 cancer cells. Results demonstrated that compounds **1k–1o** (Figure 4) were considered active against these three cell lines, which encourages the need for further and more detailed studies.

A group of investigators focused their attention on androstene oximes and their *O*-alkylated derivatives [45]. These compounds and two previously synthesized oxime derivatives, compounds **1p** [46] and **1q** [47] (Figure 4) were evaluated in leukemia, colon, melanoma, and renal cancer cell lines. Only compounds **1p** and **1q** showed considerable cytotoxicity in all cell lines (percentages of growth of 10.07 to 75.01% at 10 µM), being this effect more pronounced in the leukemia cell lines, K562, HL60, and SR.

Aiming to evaluate the combined effect of the 17-heterocyclic ring and hydroxyimino function, a group of investigators designed and synthesized modified 17α-picolyl and 17(*E*)-picolinylidene androstane derivatives and their antiproliferative activity against breast, prostate, cervical, colon and lung adenocarcinoma, as well as normal fetal lung fibroblasts was assessed [48]. MTT assay results (Table 1) demonstrated that compound **1r** (Figure 4) was the most active compound in PC3 cells (IC_50_ = 6.6 µM), while compound **1s** (Figure 4) was more active, not only against PC3 cells (IC_50_ = 8.7 µM) but also against MCF7 cells (IC_50_ = 1.7 µM). Given these encouraging results, the authors went further ahead and studied deeply the potential mechanisms of action of compound **1s** in MCF7 cells. Results showed that **1s** induced apoptosis in breast cancer cells, assessed by alterations in the cells morphology, such as nuclear condensation, vacuolated cytoplasm, degradation of nuclei and cytoplasm, membrane blebbing, and apoptotic bodies formation [48].

Savić et al., designed and synthesized some novel D-homo lactone androstane derivatives and evaluated their antiproliferative activity against several cancer cell lines [49]. Among these compounds, the oxime derivative **1t** (Figure 4), demonstrated to have high activity (Table 1). Moreover, **1t** also revealed selectivity towards cancer cells since it presents a much higher IC_50_ in the normal human cell line, MRC5. A few years later, the same group of investigators designed, synthesized and evaluated the antitumor activity of some more new D-homo lactone androstane derivatives [50]. In vitro cytotoxicity assessment against cancer cells revealed that the steroidal oxime **1u** (Figure 4) was able to decrease the proliferation of PC3 (IC_50_ = 27.94 µM) and HeLa (IC_50_ = 13.86 µm) cells (Table 1).

**Table 1 molecules-28-01690-t001:** IC_50_ values (µM) of the synthesized androstane oxime derivatives.

Cell Line	Compounds													
	1a	1b	1c	1d	1e	1f	1g	1h	1i	1j	1r	1s	1t	1u
DU145	3.9	-	-	-	-	-	-	-	-	-	-	-	-	-
HeLa		15.1	75.7	-	-	>100	>100	22.6	12.8	22.2	>100	81.8	36.0	13.9
SMMC7404	-	>200	184	-	-	-	-	-	-	-	-	-	-	-
WiDr	-	-	-	9.1	16.1	-	-	-	-	-	-	-	-	-
PC3	-	-	-	13.8	14.5	>100	57.7	77.1	-	-	6.6	8.7	36.7	27.9
HepG2	-	-	-	23.9	18.2	-	-	-	-	-	-	-	-	-
H1299	-	-	-	18.6	19.2	-	-	-	-	-	-	-	-	-
MCF7	-	-	-	-	-	41.0	44.9	>100	10.2	19.8	50.4	1.7	81.3	>100
MDA-MB-231	-	-	-	-	-	47.3	5.2	4.7	-	-	25.3	40.1	11.9	>100
HT29	-	-	-	-	-	4.4	10.6	3.3	-	-	>100	10.3	4.0	>100
A549	-	-	-	-	-	1.5	1.8	2.0	-	-	>100	56.0	-	-
MRC5	-	-	-	-	-	>100	>100	>100	-	-	>100	>100	>100	>100
CEM	-	-	-	-	-	>50	>50	30.4	-	-	-	-	-	-
G361	-	-	-	-	-	45.3	46.6	8.9	-	-	-	-	-	-
BJ	-	-	-	-	-	>50	>50	25.3	-	-	-	-	-	-
KB	-	-	-	-	-	-	-	-	26	28.5	-	-	-	-
MKN-28	-	-	-	-	-	-	-	-	18.1	36.1	-	-	-	-
Ref.	[35]	[39]		[34]		[42]			[43]		[48]		[49]	[50]

These values were obtained through different techniques such as MTT and SRB assays.

### 3.2. Estrane Derivatives

Estrogen-3-*O*-sulfamates (EMATEs) are a class of steroidal compounds, which act as irreversible inhibitors of steroid sulfatase (STS), an enzyme involved in the development of estrogen-dependent breast cancer [51]. Given this, Leese et al. decided to design and synthesize D ring-modified 2-substituted EMATEs and evaluated their in vitro anticancer activity in MCF7 cancer cells [52]. Antiproliferative activity evaluation of the steroidal oximes **2a**–**2j** (Figure 5) revealed that in general, all oximes were very effective in inhibiting MCF7 cancer cell proliferation (Table 2). These results reinforce the importance of the modifications in the 17-position of the 2-substituted EMATEs, particularly the introduction of an oxyimino group, which increased the antiproliferative activity of this class of compounds [52].

A series of novel estrone-16-oxime ethers were designed and synthesized, and their antitumor activity was evaluated in HeLa, MCF7, and A431 cancer cells [53]. From all the synthesized compounds, oxime **2k** and **2l** (Figure 5) decreased cancer cell proliferation in a more pronounced way, being HeLa cells the most susceptible to both compounds (IC_50_ = 4.41 µM for **2k** and 4.04 µM for **2l**) (Table 2). Further evaluation to characterize the mechanisms of action of these compounds revealed both of them interfered with the cell cycle at G1 phase and induced apoptosis in HeLa cells, by the activation of caspase-3. Following this study, the same authors continued their research on this topic and designed and synthesized a series of novel D-secooxime derivatives in the 13β- and 13α-methyl-estrone series [54]. After MTT assays to assess antiproliferative activity in HeLa, MCF7, A2780, and A431 cancer cell lines, compounds **2m**–**2o** (Figure 5) stand out for displaying high cytotoxicity against all cell lines and being, generally, even more active than cisplatin as it can be seen in Table 2. Furthermore, compound **2n** was selected for additional analysis in A2780 cells, namely cell cycle analysis. Results showed that **2n** induced cell cycle arrest at the S phase, which in turn might be responsible for apoptosis induction [54].

Cushman et al. investigated several estradiol analogs to improve the anticancer activity of 2-methoxyestradiol, a naturally occurring tubulin polymerization inhibitor [55]. Starting from 2-ethoxyestradiol, an analog previously synthesized by the same group, from 2-methoxyestradiol [56], two novel steroidal oximes **2p** and **2q** (Figure 5) were designed and synthesized and their antitumor activity was assessed. Results revealed that both compounds were extremely toxic to all the cell lines studied (HOP62, HCT116, SF539, UACC62, OVCAR-3, SN12C and DU145) with GI_50_ (half growth inhibition) at values ranging from 0.010 to 0.066 µM (Table 2). Furthermore, **2p** and **2q** were able to inhibit tubulin polymerization. The oxime derivatives were the most active among all compounds synthesized, which points out the importance of the oxyimino functionality.

Aiming to develop new steroidal oximes with potential application in cancer treatment, a group of scientists designed and synthesized a series of estrone oxime derivatives and evaluated them in six cancer cell lines [57]. MTT results proved that compound **2r** (Figure 5) is the most active against all cell lines, being LNCaP cells the most sensitive to this molecule (IC_50_ = 3.59 µM) (Table 2). Given this, the cytotoxicity of this steroidal oxime was mediated by a cell cycle arrest at G2/M phases accompanied by cell death by apoptosis, with evidence of condensed and fragmented nuclei. Moreover, the authors speculated that this compound might interfere with β-tubulin [57].

**Table 2 molecules-28-01690-t002:** IC_50_ values (µM) of the synthesized estrane oxime derivatives.

Cell Line	Compounds																	
	2a	2b	2c	2d	2e	2f	2g	2h	2i	2j	2k	2l	2m	2n	2o	2p	2q	2r
MCF7	5.87	6.47	0.24	0.17	0.23	0.23	1.33	5.14	0.17	>30	>30	2.6	2.6	>30	2.1	-	-	25.63
HeLa	-	-	-	-	-	-	-	-	-	-	-	-	1.2	7.1	1.7	-	-	-
A431	-	-	-	-	-	-	-	-	-	-	-	-	0.8	0.9	0.9	-	-	-
A2780	-	-	-	-	-	-	-	-	-	-	-	-	0.9	1.4	0.7	-	-	-
HOP62	-	-	-	-	-	-	-	-	-	-	-	-	-	-	-	0.017	1.2	-
HCT116	-	-	-	-	-	-	-	-	-	-	-	-	-	-	-	0.031	2.1	-
SF539																0.021	2.8	
UACC62	-	-	-	-	-	-	-	-	-	-	-	-	-	-	-	0.015	0.88	-
OVCAR3	-	-	-	-	-	-	-	-	-	-	-	-	-	-	-	0.016	5.4	-
SN12C	-	-	-	-	-	-	-	-	-	-	-	-	-	-	-	0.045	17	-
DU145	-	-	-	-	-	-	-	-	-	-	-	-	-	-	-	0.049	17	-
MDA-MB-231	-	-	-	-	-	-	-	-	-	-	-	-	-	-	-	0.010	6.5	-
MGM	-	-	-	-	-	-	-	-	-	-	-	-	-	-	-	0.066	4.2	-
T47D	-	-	-	-	-	-	-	-	-	-	-	-	-	-	-	-	-	43.45
LNCaP	-	-	-	-	-	-	-	-	-	-	-	-	-	-	-	-	-	3.59
HepaRG	-	-	-	-	-	-	-	-	-	-	-	-	-	-	-	-	-	18.35
Caco	-	-	-	-	-	-	-	-	-	-	-	-	-	-	-	-	-	24.33
NHDF	-	-	-	-	-	-	-	-	-	-	-	-	-	-	-	-	-	30.84
Ref.	[52]										[53]		[54]			[55]		[57]

These values were obtained through different techniques such as MTT assays and specific assay kits.

### 3.3. Pregnane Derivatives

Bearing in mind the importance of pregnenolone in biological systems, Choudhary et al. synthesized a series of novel pregnenolone derivatives, being that some of which were oximes [58]. After synthesis, the authors came up with compound **3a** (Figure 6) which was further evaluated in HepG2 and MDA-MB-231 cancer cells. Results demonstrated that this compound was quite active against both cell lines with IC_50_ values of 4.50 and 6.76 µM in HepG2 and MDA-MB-231 (Table 3), respectively, which makes it very promising to be further analyzed.

Chen and collaborators did their research in 4-azasteroidal derivatives, namely 4-azasteroidal-20-oxime derivatives using progesterone as starting material [10]. The compounds were then evaluated by the MTS assay for their anticancer activity in human bladder carcinoma. After the synthesis, the authors came up with five very active oxime compounds (**3b**–**3f**, Figure 6), in T24 cells (Table 3). After structure-activity relationships (SAR) analysis, Chen et al. concluded that the methyl and ethyl oxime-ether derivatives were more active against the referred cell line when compared with the aryl oxime-ester derivatives.

Using pregnenolones as precursors, a group of scientists designed and synthesized a series of benzylidene pregnenolones and their oximes and further evaluated their potential antitumor activity in several cancer cell lines, namely HT29, HCT15, SF-295, HOP62, A549 and MCF-7 [59]. From all the synthesized oximes, compounds **3g** and **3h** (Figure 6) were the most active against HCT15 (IC_50_ = 0.31 µM for **3g** and 0.65 µM for **3h**) and MCF7 (IC_50_ = 0.60 µM for **3g** and 1.91 µM for **3h**) (Table 3) cancer cells, revealing a cell specificity. Moreover, the oxime derivatives were more potent than the corresponding precursors, which points out the importance of the oxyimino functionality in conferring cytotoxicity against cancer cells.

### 3.4. Cholestane Derivatives

Krstić et al. reported the design and synthesis of two novel steroidal oximes from cholesterol, compounds **4a** and **4b** (Figure 7) [60]. A few years later, the same group of scientists decided to evaluate the potential antitumor activity of these two oximes [61] against two human cancer cell lines (HeLa and K-562) and against non-stimulated and PHA-stimulated peripheral blood mononuclear cells (PBMC’s) from healthy donors to assess selectivity. Both compounds showed a dose-dependent decrease in the proliferation of HeLa cancer cells (IC_50_ = 35.24 ± 4.09 for **4a** and IC_50_ = 20.68 ± 3.10 for **4b**) and K-562 cancer cells (IC_50_ = 28.05 ± 10.18 for **4a** and IC_50_ = 11.16 ± 1.24 for **4b**), while showing almost no effects in normal immunocompetent cells (Table 4). Further evaluation revealed that compound **4b** exerted its cytotoxicity by inducing apoptosis in HeLa cells. On the contrary, compound **4a** showed no evidence of apoptosis or necrosis, which can be explained by the difference in the *Z/E* stereochemistry in the hydroxyimino group of both compounds [61].

A group of researchers investigated in aza-homosteroids derived from diosgenin and cholesterol-containing hydroxyimino and lactam groups in the A/B ring with four types of side chains: cholestane, spirostane, 22-oxocholestane and 22,26-epoxycholestene [62]. These compounds were further evaluated as potential antitumor agents in MCF7 cancer cells. Antitumor activity assessment revealed that from all the synthesized compounds, oximes **4c**–**4e** (Figure 7) were the most promising ones with an IC_50_ of 8.2, 9.5, and 7.9 µM, respectively (Table 4). MCF7 cancer cells were more sensitive to the compounds containing a hydroxyimino group in comparison with the compounds without this chemical function. Moreover, Mora-Medina et al., evaluated these three oximes against PBMCs to test for the selectivity index of these compounds. Results showed that **4c-4e** were all remarkably selective for MCF-7 cells, which makes these compounds very promising [62].

Huang et al. synthesized a series of 6-hydroxyimino-substituted-3-aza- and 4-aza-A-homo-3-oxycholestanes using cholesterol as starting material [63]. Oximes **4f** and **4g** (Figure 7) were further evaluated in three different cancer cell lines, namely HeLa, SMMC7404, and MGC7901. Results showed that both compounds displayed cytotoxicity against these cell lines (Table 4), being more active than the referenced drug, cisplatin, in HeLa and SMMC7404 cells. Altogether, these results demonstrated that the introduction of the hydroxyimino group at position 6 was crucial for the compounds’ cytotoxicity against cancer cells. Given the good outcomes obtained, these investigators decided to further analyze compound **4f** (Figure 7) and evaluated its antitumor activity in five more cancer cell lines (GNE2, SPC-A, Tu686, PC3, and HT29) [64]. Results indicate that oxime **4f** was also quite active in all cell lines with IC_50_ values ranging from 10.6 to 74.5 µM (Table 4). Furthermore, the molecular mechanisms by which **4f** decreased cancer cell proliferation were studied. Results unveiled that compound **4f** induced cancer cell apoptosis by activation of the intrinsic pathway, which was demonstrated by the annexin V labeling, activation of caspase-3, and release of cytochrome C. Moreover, this oxime was also evaluated in an in vivo model and proved to be able to inhibit tumor growth [64].

Oxysterols are a group of lipids derived from cholesterol, particularly interesting in the medicinal chemistry field due to their diverse biological effects [65]. Given this, Carvalho et al. designed and synthesized a series of oxysterols in which oxime **4h** (Figure 7), a 3β,5α,6β-trihydroxycholestanol derivative, was included [66]. After the synthesis of **4h**, the compound’s cytotoxicity was studied in five human cancer cell lines (HT29, SH-SY5Y, HepG2, A549, PC3) and two human normal cell lines (ARPE-19 and BJ). Oxime **4h** was quite active in all cancer cell lines being that HT29 was the most sensitive (IC_50_ = 11.9 µM). Moreover, the compound revealed some selectivity towards HT29 cells (selectivity index of 1.99) since the IC_50_ displayed was higher in the normal cell lines (Table 4). This work contributed to deepening the understanding of oxysterols’ cytotoxicity and shed some light on the SAR of these classes of compounds.

In 1997, two steroidal molecules with very interesting and unusual structures, (6*E*)- hydroxyiminocholest-4-en-3-one (**4i**, Figure 7) and its 24-ethyl analog (**4j**, Figure 7), were isolated from *Cinachyrella* marine sponges [67]. Analysis of their antitumor activity revealed that only compound **4i** was able to decrease the proliferation of P388 (IC_50_ = 1.25 µg/mL), A549 (IC_50_ = 1.25 µg/mL), HT29 (IC_50_ = 1.25 µg/mL), and MEL28 (IC_50_ = 2.5 µg/mL) cells (Table 5). Following this discovery, Deive et al. [68] further explored the SAR of this type of compound and prepared several derivatives of **4i** and **4j** with different structural features, namely with different side chains and degrees of unsaturation on ring A. From all the synthesized compounds, **4k**–**4o** (Figure 7) stood out with an IC_50_ ranging from 0.125 to 1.25 µg/mL (Table 5) in the above-mentioned cancer cell lines. These results allowed to shed some light regarding SAR analysis and demonstrated that the presence of a cholesterol-type side chain, a ketone group at C-3 and a high degree of oxidation in ring A might play a major role in the compounds’ cytotoxicity [68]. A few years later, the same group continued their investigation in 6-hydroxyiminosteroids [69] and, bearing in mind the information about the SAR of these compounds, a series of new steroidal oximes were synthesized and evaluated as potential antitumor agents. After biological evaluation in four different cancer cell lines (A549, HCT116, PSN1, T98G), compounds **4p**–**4w** (Figure 7) were the most effective in decreasing cancer cell proliferation, demonstrating good IC_50_ values (Table 4). Once again, the oxygenation of ring A turned out to be very important in the increased cytotoxicity of the compounds against cancer cells [69].

Another study by Cui et al. [70] also reported the synthesis and further biological evaluation of 6-hydroxyiminosteroidal cholestane derivatives. They not only developed a facile and efficient synthetic method for the synthesis of the natural compounds **4i** and **4j** (Figure 7) but also designed and synthesized a new steroidal oxime (**4x**, Figure 7). Further antitumor activity evaluation in four human cancer cell lines, namely Sk-Hep-1, H292, PC3, and Hey1B revealed that compound **4x** presented modest cytotoxicity against these cell lines with IC_50_ values ranging from 37 to 59.5 µg/mL (Table 5). Once again and compared to the cytotoxicity displayed by compounds **4i** and **4j**, the structure of the side chains impacts the cytotoxicity of the compounds. The cholesterol-type side chain seems to be important for biological activity, which goes accordingly to the results obtained by other research groups [68]. The same group of investigators continued their research on this topic and synthesized more steroidal oximes with the hydroxyimino groups in different locations (A ring or B ring) and with different types of side chains at position 17 [71]. From all the derivatives, compounds **4y**–**4ag** (Figure 7) were the ones with the best antitumor activity against Sk-Hep-1, H292, PC3, and Hey1B (Table 5). This study reinforced the conclusions obtained in the previous ones [68,69,70], where it is stated that for enhanced cytotoxicity the compounds must have a cholesterol-type side chain, a hydroxyimino group on the B ring, and a hydroxy group on the A or B ring. The same authors went further ahead in the SAR and synthesized a series of derivatives similar to the ones synthesized in the previous study but without the 4,5-double bond [72]. After biological activity evaluation, compounds **4ah**–**4ak** (Figure 7) proved to be the most effective in decreasing Sk-Hep-1, H292, PC3, and Hey1B cancer cells proliferation with IC_50_ values ranging from 35.4 to 103 µg/mL (Table 5). These compounds (without the 4,5-double bond) were more active than the equivalent ones with the 4,5-double bond [71], which indicates that the double bond in this particular position confers a negative effect in the biological activity displayed by these derivatives. All these studies were very important in unraveling the SAR of 6-hydroxyiminosteroids and might help shed some light on the design of novel chemotherapeutic drugs for the treatment of different types of cancer.

Gan et al. designed and synthesized some steroidal hydrazone derivatives with 3,6-disubstituted structure and different side chains at 17-position, being the hydroxyimino group one of these substitutions, namely in the 3-position [73]. Antiproliferative activity evaluation was carried out in vitro against gastric and liver cancer cells after 72 h of incubation. The synthetic routes gave rise to a series of novel molecules among them, compounds **4al**–**4an** (Figure 7), which were remarkably active against the two cancer cell lines used, Bel7404 and SGC7901 (Table 6). Compound **4am** was particularly active to Bel7404 cells (IC_50_ = 7.4 µM) being three times more active than cisplatin (IC_50_ = 22.3 µM), which makes **4am** a potential antitumor drug.

Huang et al. published a study describing the synthesis of new sulfated hydroxyiminosterols as potential antitumor agents [74]. In vitro antiproliferative activity, assessed in HeLa, SMMC7404 and MGC7901 cancer cell lines revealed that compounds **4ao**–**4ar** (Figure 7) were all able to decrease all cancer cell lines proliferation (Table 6). Please note that compound **4ao** was even more cytotoxic than cisplatin against all the cancer cell lines studied.

A group of investigators from Argentina synthesized three new 6*E*-hydroxyiminosteroids and then assessed their antitumor activity against prostate cancer cells (PC3 and LNCaP) [75]. After antiproliferative activity evaluation, compounds **4as**–**4av** (Figure 7) proved to be quite active in both cancer cell lines with IC_50_ values ranging from 10.8 to 44.8 µM (Table 6).

Using analogs of compounds **4i** and **4j** as precursors, Huang et al. designed and synthesized novel steroidal oximes and then evaluated their anticancer efficacy against a panel of six cancer cell lines [76]. Of all the synthesized compounds, oxime **4aw** (Figure 7) was the most powerful oxime, especially in HeLa and GNE2 cancer cell lines with IC_50_ values of 9.1 and 11.3 µM, respectively (Table 6). Remarkably, this compound was even more active than cisplatin (IC_50_ = 10.1 and 16.8 µM in HeLa and GNE2 cells, respectively).

### 3.5. Diosgenin Derivatives

Diosgenin is a natural steroidal sapogenin, which was first isolated from *Discorea tokoro* by Takeo Tsukamato in 1936 [77]. This molecule is widely used in the pharmaceutical industry as the main precursor in the synthesis of steroids. Given this, several investigators have been focusing their attention on diosgenin derivatives. Sánchez et al. designed and synthesized two novel oxime derivatives (**5a** and **5b**, Figure 8) using diosgenin as a precursor and then tested their antitumor activity against HeLa and CaSki cancer cell lines [78]. Results revealed that both compounds caused a dose-dependent decrease in HeLa and CaSki cell proliferation with IC_50_ ranging from 10.9 to 48.18 µM (Table 7) and compound **5a** was even more active than diosgenin, which reinforces the importance of the hydroxyimino group, positioned in the side chain, in conferring cytotoxicity. Further analysis of both compounds demonstrated that they exerted their antitumor activity by interfering with the cell cycle, and causing cell death by apoptosis mediated by the activation of caspase-3, which, in turn, is responsible for DNA fragmentation [78].

Another group of investigators synthesized a series of diosgenin derivatives by introducing new modifications in rings A and B [79]. Among all these compounds, oxime **5c** (Figure 8) was evaluated in HeLa, MDA-MB-231, and HCT-15 cancer cells to assess its antitumor activity. Results showed that **5c** was quite active in all cell lines with IC_50_ of 18.23, 10.83, and 17.56 µg/mL in HeLa, MDA-MB-231, and HCT-15 (Table 7), respectively. Furthermore, this compound was not toxic to PBMC which can point to a selectivity towards cancer cells.

Carballo et al. designed and synthesized a series of novel hydroxyimino steroidal derivatives and evaluated their antiproliferative activity in MCF7 and MDA-MB-231 breast cancer cells [80]. Compounds **5d**–**5e** (Figure 8) were able to decrease cell proliferation with IC_50_ values ranging from 9.3 to 11.8 µM (Table 7). Interestingly, compounds **5d** and **5e** (spirostan derivatives) were more active against the triple-negative cells, a subtype of breast cancer associated with a worse prognosis and fewer therapeutic options.

More recently, a group of scientists designed and synthesized a series of steroidal oximes from diosgenin and evaluated their antitumor activity against six cancer cell lines (A549, HBL100, HeLa, SW1573, T47D, and WiDr) [81]. Of all the compounds synthesized, **5f** and **5g** (Figure 8) were the most active against all cancer cell lines tested (Table 7). Moreover, compound **5f** was more active than cisplatin in T47D and WiDr cells and **5g** was more active only in WiDr cells. These results encourage more research to unravel the mechanisms of action behind the cytotoxicity displayed by these compounds against cancer cells.

### 3.6. Bile Acids Derivatives

Bile acids have shown good biological and chemical properties, which make them very useful in the designing of new pharmacological entities [82]. Given this, a group of investigators reported the synthesis of new 3-aza-A-homo-4-one bile acid and 7-deoxycholic acid derivatives, among them some with the hydroxyimino group in their structure [83]. Of all the compounds, **6a** and **6b** (Figure 9) were the most promising ones being very effective in decreasing the MGC7901, HeLa, and SMMC7404 cells proliferation. Please note that the most sensitive cell line to both **6a** and **6b** was the HeLa, ovarian cancer cell line, with an IC_50_ of 14.3 and 24.3 µM, respectively. Compound **6a** was even more cytotoxic against cancer cells than the reference drug, cisplatin (IC_50_ = 20.6 µM). These results enlighten, once again, the relationship between the hydroxyimino group and biological activity [83].

## 4. Steroidal Oximes as Antimicrobial Agents

Infectious diseases are a public health problem and are among the top ten leading causes of death worldwide according to WHO [84]. The need for novel therapeutic options to treat this type of disease is of great importance. Steroids and hydroxyimino group-bearing compounds have been shown to be effective against bacteria, fungi, and some viruses [12,85]. In this section, we will describe the most potent steroidal oximes with antibacterial, antifungal, and antiviral activity.

### 4.1. Androstane Derivatives

A series of androstane derivatives containing the hydroxyimino group (**7a**–**7e**, Figure 10) were designed, synthesized, and further analyzed as antibacterial and antifungal drug candidates [86] against different strains of bacteria and fungi. Results demonstrated that all oximes presented excellent antibacterial and antifungal activity, being in general more toxic to the pathogens than the referenced drugs, as seen by the minimal inhibitory (MIC) and minimal bactericidal/fungicidal (MBC/MFC) concentrations described in Table 8 and Table 9.

### 4.2. Pregnane Derivatives

Using pregnenolone as starting material, Prabpayak et al. synthesized a novel oxime derivative (**8a**, Figure 11) and evaluated its antibacterial activity against a series of different strains of bacteria [87]. Results revealed that **8a** was able to decrease bacterial growth, as seen by the calculation of the zone of inhibition which was 12.5 mm for *S. mutans* ATCC 1275 and 25 mm for *C. diptheriae*.

Lone et al. designed and synthesized nine oximes of steroidal chalcones (**8b**–**8k**, Figure 11) and screened them for in vitro antimicrobial activity against different strains of bacteria and fungi [88]. All compounds displayed good antimicrobial activity against all strains studied (Table 10). Moreover, the oximes showed enhanced antimicrobial activity when compared with the corresponding chalcones with a carbonyl group instead of a hydroxyimino group. These results highlight the importance of this hydroxyimino group for the toxicity displayed against pathogens.

### 4.3. Cholestane Derivatives

A group of scientists designed and synthesized three novel steroidal oximes (**9a**–**9c**, Figure 12) from cholestane [89]. After synthesis, the antibacterial and antifungal activity in different strains of bacteria (*S. pyogenes*, *S. aureus*, *S. typhi*, *P. aeruginosa* and *E. coli*) and fungi (*P. marneffei*, *A. fumigatus*, *T. mentagrophytes*, *C. albicans* and *C. krusei*) was assessed through analysis of the diameter of zone of inhibition (mm). Anthelmintic activity was also analyzed against earthworms. Generally, all compounds presented good antibacterial and antifungal activity with compound **9c** being the most active against both bacterial (zone of inhibition ranging from 17.3 to 23.9 mm) and fungi strains (zone of inhibition ranging from 15.1 and 25.5 mm). This compound was also the most effective anthelmintic compound showing great early paralysis and lethal times. Further docking studies demonstrated that **9c** had not only better affinity to the receptor, but also presented the best docking score, making it a very promising antimicrobial [89].

Oxysterols apart from displaying antitumor activity have also been associated with antimicrobial activity [90]. Compounds **9d**–**9g** (Figure 12) were designed and synthesized and further evaluated against a series of fungal strains [91]. In general, all compounds displayed good antifungal activity against all the strains used in this study (MIC values ranging from 2 to 64 µg/mL, Table 11). *C. neoformans* (CN1) was particularly sensitive to all oximes synthesized, being the MIC values (2–4 µg/mL) presented even lower than the referenced drugs fluconazole (16 µg/mL) and amphotericin B (32 µg/mL).

Aiming to develop novel therapeutic options for the treatment of herpes simplex virus, Pujol et al. synthesized a series of sulfated steroids derived from 5α-cholestanes [92]. The oxime **9h** (Figure 12) was the most promising compound, showing the best inhibitory values of HSV-1, HSV-2, and pseudorabies virus (PrV) strains, including acyclovir-resistant strains in human and monkey cell lines (EC_50_ values ranging from 16.7 to 25 µg/mL, Table 11). Moreover, the authors went further ahead and decided to investigate the mechanism of action of **9h** in HSV-1. After using the virucidal assay, authors concluded that the **9h** was not able to affect the initial steps of virus entry but rather inhibited an ensuing event in the infection process of this virus [92].

### 4.4. Bile Acids Derivatives

Hepatitis B is a major global health problem and is caused by the hepatitis B virus (HBV). This condition affects the liver, leading to cirrhosis and liver cancer, which can lead to patient death [93]. In this context, a group of scientists focused their attention on designing and synthesizing oxime derivatives of dehydrocholic acid as potential HBV drugs [94]. After anti-hepatitis B virus activity evaluation in HepG 2.2.15 cells, compounds **10a**–**10c** (Figure 13) exhibited more cytotoxicity against the virus than the referenced drug, entecavir, as seen by the most effective inhibition of HBeAg (hepatitis B -antigen). Compound **10b** was the most active compound showing significant anti-HBV activity on inhibition secretion of HBeAg (IC_50_ = 96.64 µM) when compared, once again, with entecavir (IC_50_ = 161.24 µM). Moreover, docking studies were also performed to evaluate the potential mechanisms behind the activity of these compounds. Results point out a possible interaction with protein residues of heparan sulfate proteoglycan (HSPG) in host hepatocytes and bile acid receptors [94].

## 5. Steroidal Oximes as Anti-Inflammatory Agents

### 5.1. Cholestane Derivatives

Since inflammation plays a critical role in tumor progression, Díaz et al. designed and synthesized a series of 22-oxocholestane oximes as potential anti-inflammatory agents in an acute inflammation mouse ear model [95]. Five oxime derivatives (**11a**–**11e**, Figure 14) stood out from the rest as being the most promising, being able to reduce ear inflammation and edema. Moreover, these compounds also repressed the expression of pro-inflammatory genes such as TNF-α, COX-2, and IL-6, making them very promising lead candidates for further assessment.

### 5.2. Bile Acid Derivatives

The need for anti-inflammatory drugs such as glucocorticoids has been increasing, especially since the COVID-19 pandemic, because this type of compounds is the standard treatment for this disease. Bearing this in mind, Bjedov and collaborators designed, synthesized, and screened the binding activity of a series of bile acid derivatives for the ligand-binding domain of glucocorticoid receptor (GR-LBD), the main receptor in the anti-inflammatory process [96]. Of all the synthesized compounds, oxime **12a** (Figure 15) presented the best relative binding affinity for GR-LBD. Even though the authors were not able to perform molecular docking and predict the interactions between compound **12a** and the enzyme, they suggest that this binding affinity could be attributed not only to the C-24 carboxylic group but also to the hydroxyimino groups. Moreover, the C-12 hydroxyimino group could also be used as an alternative hydrogen donor to the 11β-OH group of the glucocorticoids and the hydroxyimino group at C-3 may also be beneficial for glucocorticoid receptor affinity through the establishment of a hydrogen bond. This study was very important and helped shed some light on the possible interactions of steroidal oximes with their targets.

## 6. Steroidal Oximes in Clinical Development

### Istaroxime

Istaroxime (Figure 16) is an inhibitor of sodium/potassium adenosine triphosphatase (Na+/K+ ATPase), which is currently under phase 2 clinical trial development for the treatment of acute decompensated heart failure [97,98]. Apart from this, istaroxime has also been evaluated as a potential antitumor agent in several types of cancer [99,100].

## 7. Steroidal Oximes in Clinical Use

### 7.1. Norgestimate

Norgestimate (Figure 17), a steroidal oxime, (brand names, Ortho Tri-Cyclen or, Previfem, among others) is a progestin, which is used in hormonal contraception and menopausal hormone therapy for the treatment of some menopausal symptoms [101]. Norgestimate is not available as a single therapy, being used together with ethinylestradiol in birth control pills and in combination with estradiol in menopausal hormone therapy [102]. It was first introduced in the USA in 1999 and is one of the most prescribed birth control pills worldwide. This compound is sold as a mixture of the two *E* and *Z* conformers.

This oxime has been subject to further biological evaluation such as antibacterial assessment and proved to be a promising lead compound to treat biofilm-associated infections and to resensitize bacterial strains resistant to some antibiotics [103].

### 7.2. Norelgestromin

Norelgestromin (Figure 18), or norelgestromine (brand names, Evra or Ortho Evra, among others) is also a progestin used for birth control. Like norgestimate, norelgestromin is not available as a single drug but rather in combination with an estrogen, ethinyl estradiol in the form of contraceptive patches [104]. It was first introduced to the market in 2002. This compound is also sold as a mixture of two *E* and *Z* isomers.

## 8. Mechanisms of Action of Steroidal Oximes

Throughout this review, we came upon a wide variety of synthesized steroidal oximes, starting from androstane to estrane, pregnane, cholestane, diosgenin, and bile acids derivatives with very different mechanisms of action.

Most of the compounds summarized here are being evaluated as potential antitumor agents and exert their cytotoxicity against cancer cells mainly by inducing apoptosis; however, the pathways leading to it differ from compounds. For example, the androstane derivatives appear to induce apoptosis [34,42,48] by cell cycle arrest at different phases and increased ROS production [34]. Necroptosis might also be a mechanism involved in the compounds’ cytotoxicity against cancer cells [34]. Oxime estrane derivatives induced cell death by apoptosis through cell cycle interference at G1 [53], S [54] and G2/M phases [57] and through activation of caspase-3 [53]. This class of compounds also interferes with microtubules by interfering with β-tubulin [55,57]. These are often related to cell cycle arrest and, consequently, a decrease in cell division and cell death. Concerning oxime pregnane, diosgenin, and some cholestane derivatives, they cause cell cycle alterations, and cell death by the activation of the apoptotic intrinsic pathway mediated by caspase-3 activation [64,78] and release of cytochrome C [61,64,78]. Additionally, and since inflammation plays an important role in carcinogenesis, inhibition of pro-inflammatory genes such as TNF-α, COX, and IL6 is also another important mechanism displayed by these steroidal oximes [95]. Additionally, for some of the cholestane derivatives, their mechanism of action is not well elucidated since most of the studies were conducted mainly to infer about SAR, rather than go deeper into their mode of action. However, the breakthroughs reached in this field are of great importance and encourage the need to further analyze their mechanisms of action.

Aside from antitumor activity, steroidal oximes also display antimicrobial activity. However, little is known about the mode of action of these compounds. Regarding antiviral activity, it seems that steroidal oximes exert their effect by inhibition of the infection process of the virus after it enters the cells [92] and by a possible interaction with protein residues of heparan sulfate proteoglycan (HSPG) in host hepatocyte and bile acid receptor [94].

Concerning the compounds in clinical trials and already in clinical use, their mechanisms are very different. Istaroxime is an inhibitor of the Na^+^/K^+^ ATPase pump [98], whereas norgestimate and norelgestromin, which are already in the market for birth control, act as progesterone agonists, inhibiting ovulation [101,102].

## 9. Structure-Activity Relationships of Steroidal Oximes

The biological activity of a compound is closely related to its chemical structure. Steroidal oximes exert several types of biological activities, which can vary depending on the position of the oxime functionality on the steroid scaffold. Given this, some SAR can be elucidated especially for the antitumor and antimicrobial activity to help in the design and synthesis of novel molecules with pharmaceutical potential (Figure 19, Figure 20 and Figure 21). However, the conclusions that can be drawn from Figure 19, Figure 20 and Figure 21 must be analyzed carefully, as they are just trends that were observed from the compounds studied in this review article, which were the most active compounds of each paper, and were evaluated with different methodologies, different times, and in different cancer cells.

Regarding the antitumor activity, when comparing the different IC_50_ values displayed by the androstane oxime derivatives, and despite the presence of other functional groups, it seems that position C-3 of the steroidal scaffold is preferable since the compounds with a hydroxyimino group in this position were, in general, the ones with the lower IC_50_ values (**1f**, **1g**, **1h**). Compound **1h**, which has a substituted oxyimino group at position C-3 was generally the compound with the best IC_50_ values in all cell lines reported in Table 1. Interestingly, the compounds that, in addition to having a hydroxyimino group at position C-3, also had another hydroxyimino group at position C-6 (**1b**, **1c** and **1u**) or at position C-17 (**1k**–**1o**) were slightly less active, suggesting that only a single hydroxyimino group at position C-3 is preferable and might be enough for the antitumor activity. Furthermore, in some types of cancer, namely in breast and prostate cancers, a hydroxyimino group at position C-2 was also important for the cytotoxicity displayed against cancer cells (compound **1r** and **1s**). Compound **1s** combines another hydroxyimino group at position C-4, which proved to be better than with just a single hydroxyimino group at position C-2. Positions C-17 and C-7 seem to be less favorable than the other positions mentioned above. Regarding the estrane series, position C-6 seems to be the most preferable concerning antitumor activity since compounds **2p** and **2q** were the ones with best IC_50_ values. In general, compounds with a hydroxyimino group at position C-17 were also very active. Concerning the pregnane series, all oximes (substituted or not) present high activity with very low IC_50_ values (IC_50_ values ranging from 0.31 to 7.17 µM. A hydroxyimino group at position C-6 appears to be the most advantageous position, regarding the cholestane series, since they present the lowers IC50s. With regard to the diosgenin series, the compounds with just a single hydroxyimino group were slightly more cytoxic against cancer cells than the ones containing two hydroxyimino groups (compounds **5b**, **5f** and **5g** present higher IC_50_ values than compounds **5a**, **5d** and **5e**). Finally, introduction of a hydroxyimino group in the B ring was clearly more beneficial than at C-ring, considering that compound **6a** presented a lower IC_50_ value than compound **6b**, for the case of bile acid derivatives.

Considering the antimicrobial activity, the androstane oxime derivatives with just a single hydroxyimino group were slightly more potent than the compounds with two hydroxyimino groups, suggesting that a single hydroxyimino group is better for antimicrobial activity rather than two hydroxyimino groups. Furthermore, a hydroxyimino group at position C-7 might be a better option when designing and synthesizing oximes in the cholestane series, since the compounds with a hydroxyimino group at this position were slightly more toxic against pathogens than compounds with a hydroxyimino group at position C-6. Pregnane oxime derivatives were more active than the parent pregnane compounds.

To sum up, for the antitumor activity, SAR analysis points out that the pregnane scaffold might be the best option when designing novel molecules with antitumor activity. This steroidal oxime series presented the best IC_50_ values against the cancer cell lines studied when compared with the other above-mentioned series. As for the antimicrobial activity, and since the number of studies is considerably less, it turns out to be more difficult to establish robust SAR, but it seems that the androstane scaffold may be the most favorable option. In Figure 19 we can see an overview of the most common SAR of steroidal oximes for the antitumor and antimicrobial biological activities.

As mentioned at the beginning of this chapter, SAR are extremely import for the design and synthesis of novel drugs. Given this, it is our hope that this review might help guide researchers to design more potent and selective chemical entities and to help them understand the pharmaceutical potential of steroidal oximes.

## 10. Conclusions

This review sums up the work that has been developed regarding the most promising steroidal oximes in recent years. The oxyimino group is a privileged chemical function being, for that reason, very popular among the medicinal chemistry community.

Most steroidal oximes in preclinical settings present antitumor activity. In vitro and in vivo studies have demonstrated the significant antiproliferative activity and elucidated some of the mechanisms by which steroidal oximes display cytotoxicity against cancer cells, which include induction of cell cycle arrest, apoptosis, necroptosis, increased ROS production and interference with microtubules. The described targets for steroidal oximes exerting antitumor activity are caspase-3 and β-tubulin. Moreover, steroidal oximes also exert antimicrobial activity, sometimes by interfering with HSPG, resulting in tremendous cytotoxicity against bacteria, fungi, and viruses. The anti-inflammatory activity is associated with TNF-α, COX-2, and IL-6 genes expression. Finally, there are currently two steroidal oximes in the market used as birth control drugs and one in clinical trials for acute heart failure, acting by progesterone agonists and Na+/K+ ATPase pump inhibitors, respectively.

The acquired knowledge about the diversified biological activities of steroidal oximes only emphasizes the need to deeper understand the mechanisms behind their activity and consequently, unravel which are their targets. In addition, molecular interactions with targets need to be elucidated, robust SAR must be constructed, and ADME properties have to be studied in order to design more potent and selective compounds. Steroidal oximes can be, in fact, very promising lead compounds for the development of drugs for the treatment of several diseases and deserve to be better explored.

## Figures and Tables

**Figure 1 molecules-28-01690-f001:**
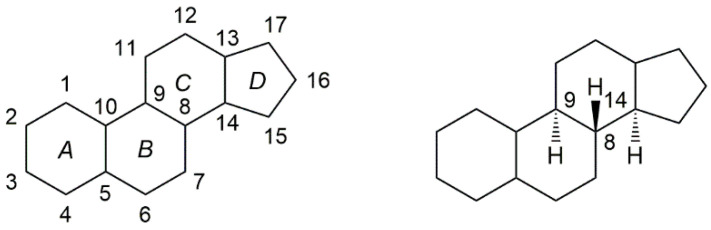
General structure of steroidal scaffold (left). Unless implied or stated to the contrary, the configuration of hydrogen atoms at the bridgehead positions 8, 9 and 14 are oriented as shown in the right formula (i.e., 8β, 9α, 14α).

**Figure 2 molecules-28-01690-f002:**
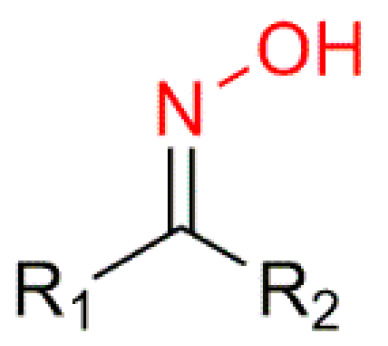
General structure of the hydroxyimino group of the oximes.

**Figure 3 molecules-28-01690-f003:**
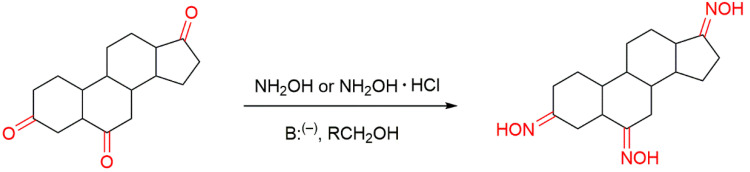
Classical synthesis of steroidal oximes which can take part in different positions in the steroidal scaffold.

**Figure 4 molecules-28-01690-f004:**
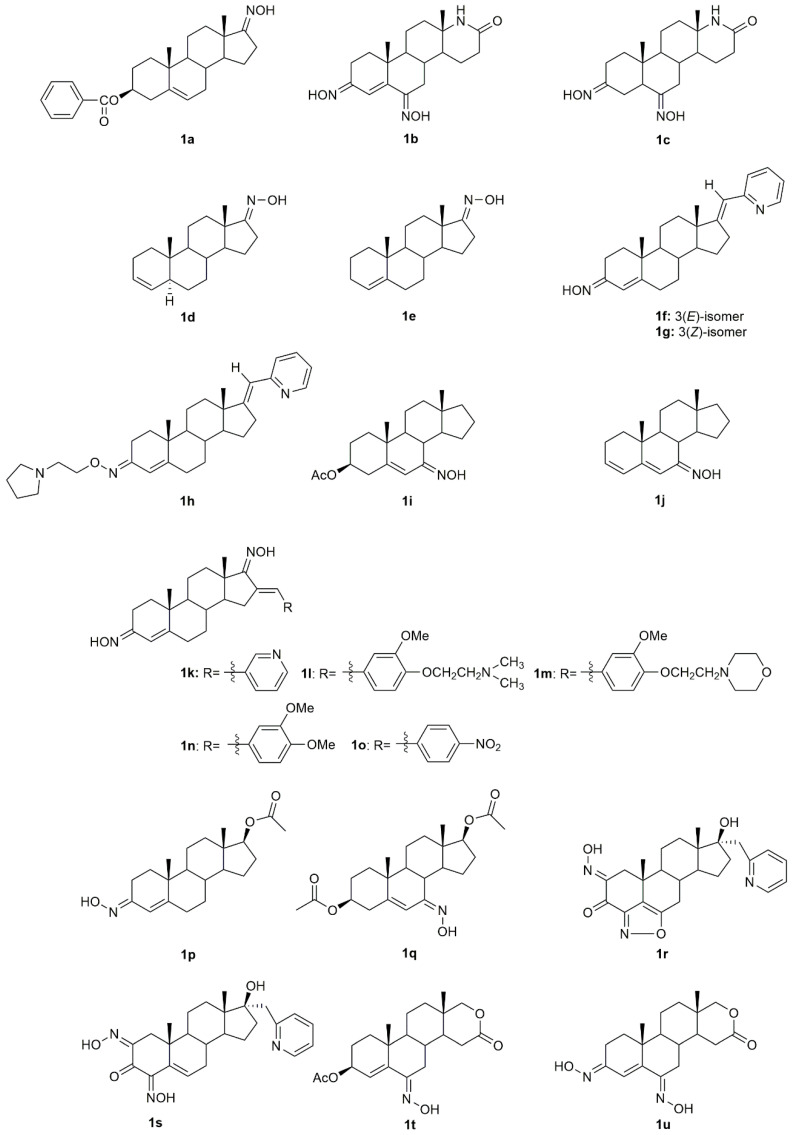
Androstane oxime derivatives with anticancer activity in several types of cancer cells.

**Figure 5 molecules-28-01690-f005:**
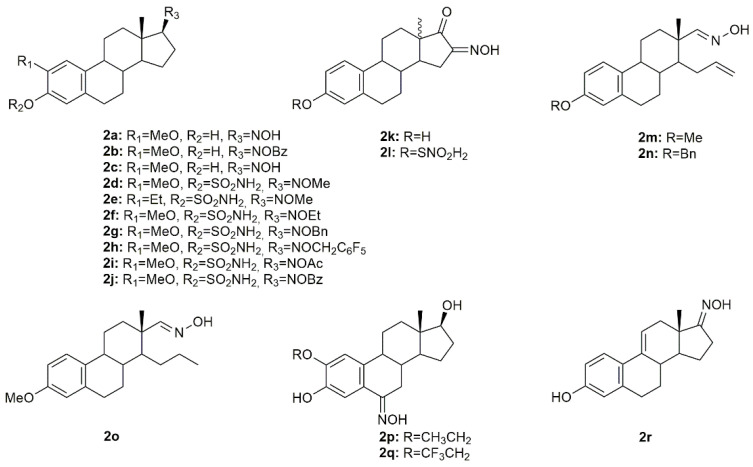
Steroidal estrane oxime derivatives with anticancer activity in several types of cancer cells.

**Figure 6 molecules-28-01690-f006:**
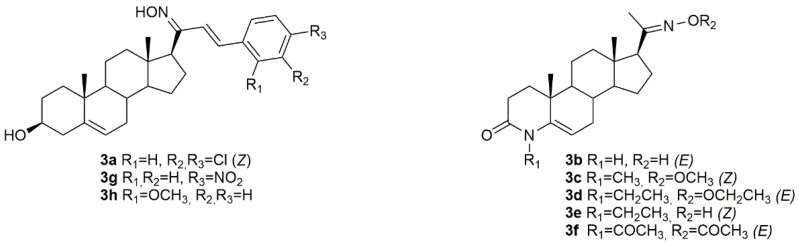
Steroidal pregnane oxime derivatives with antitumor activity in several types of cancer cells.

**Figure 7 molecules-28-01690-f007:**
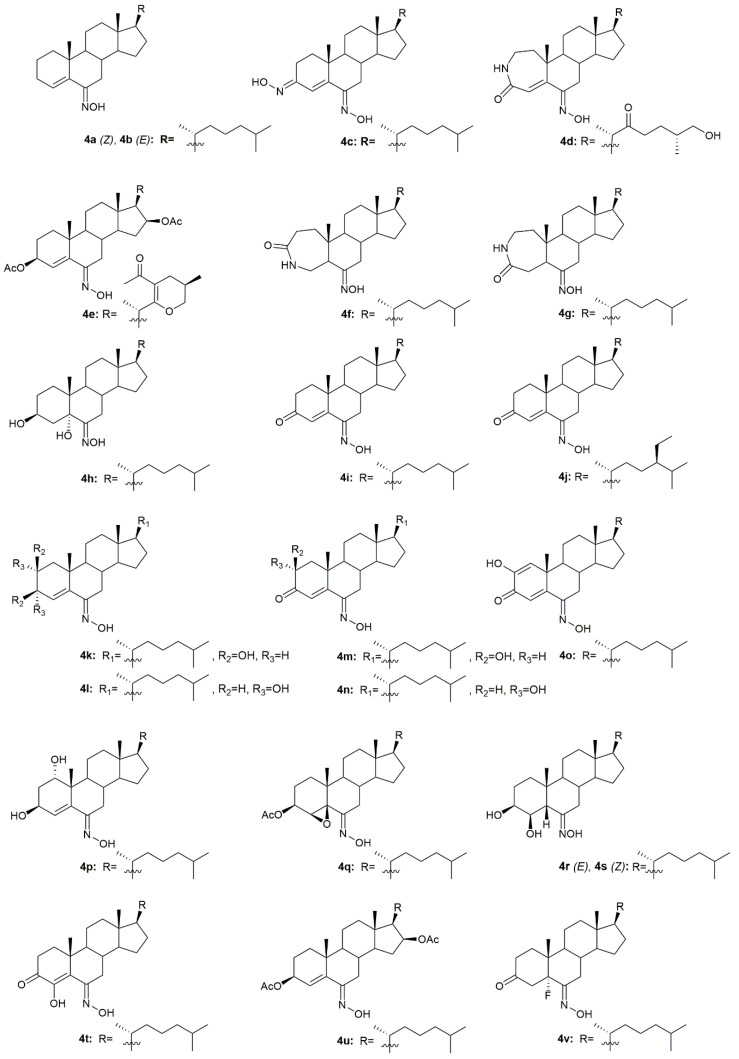
Steroidal cholestane oxime derivatives with antitumor activity in several types of cancer cells.

**Figure 8 molecules-28-01690-f008:**
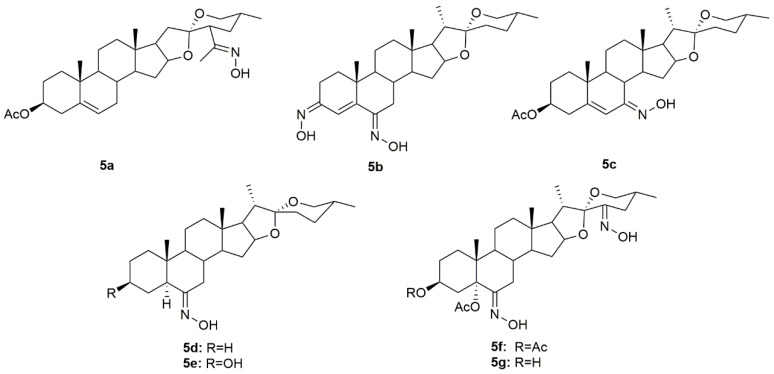
Steroidal diosgenin oxime derivatives with antitumor activity in several types of cancer cells.

**Figure 9 molecules-28-01690-f009:**
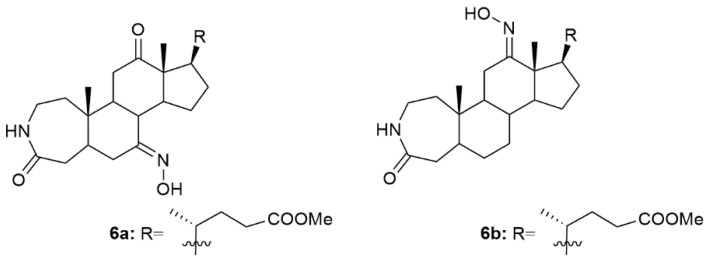
Steroidal bile acid oxime derivatives with antitumor activity in several types of cancer cells.

**Figure 10 molecules-28-01690-f010:**
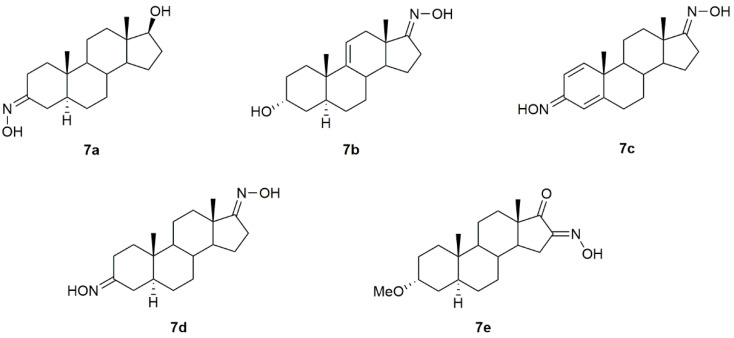
Steroidal androstane oxime derivatives with antibacterial and antifungal activity.

**Figure 11 molecules-28-01690-f011:**
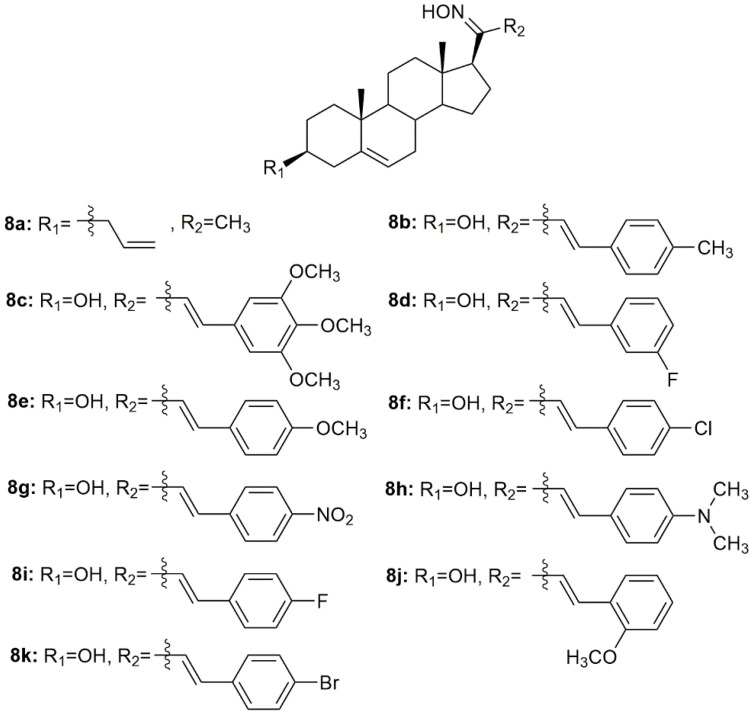
Pregnane oxime derivatives with antimicrobial activity.

**Figure 12 molecules-28-01690-f012:**
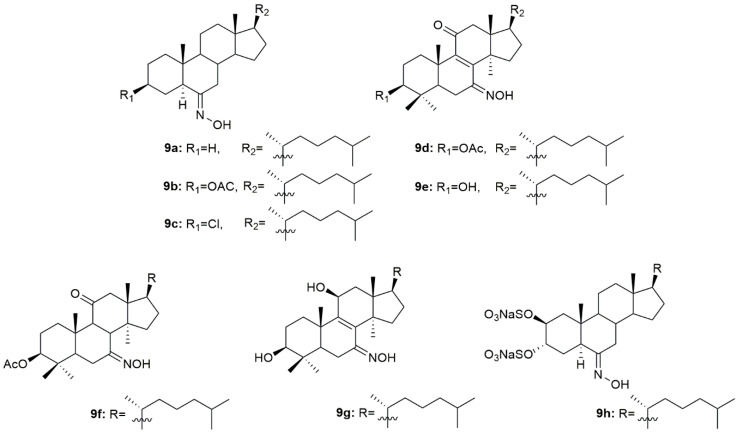
Cholestane oxime derivatives with antimicrobial activity.

**Figure 13 molecules-28-01690-f013:**
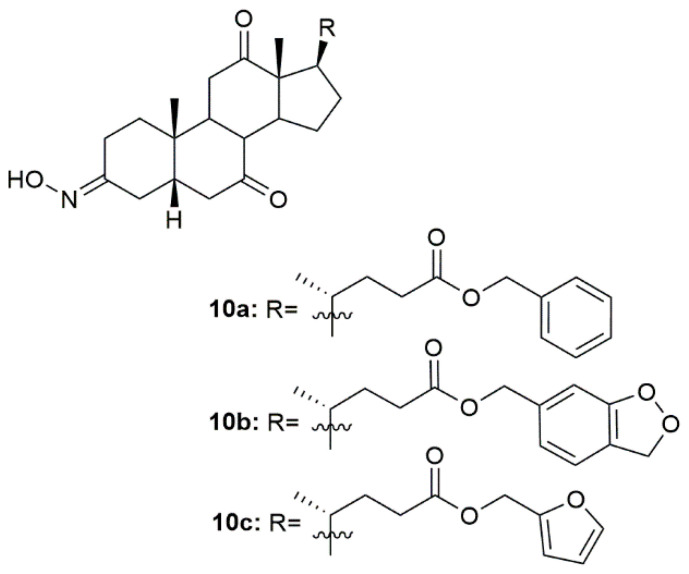
Bile acid oxime derivatives with antiviral activity.

**Figure 14 molecules-28-01690-f014:**
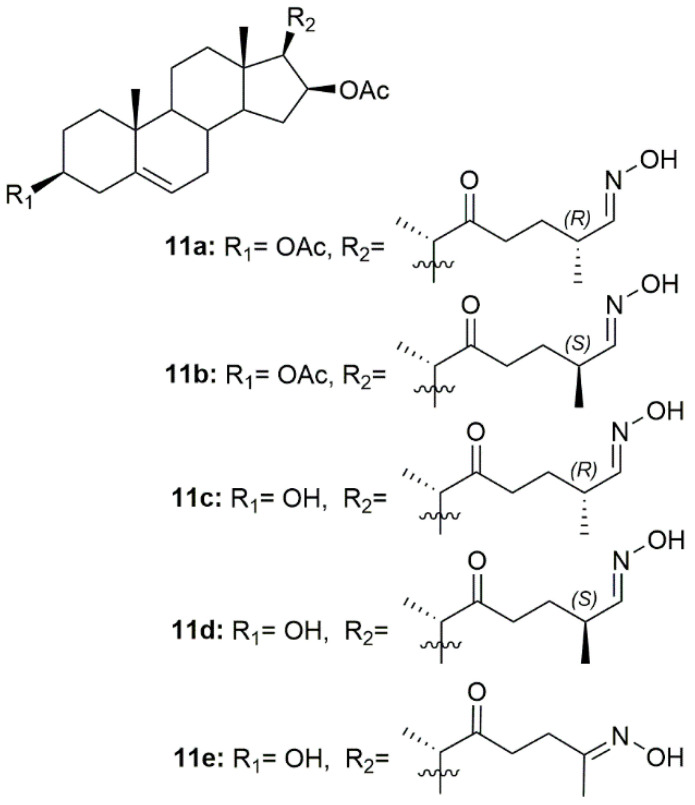
Cholestane oxime derivatives with anti-inflammatory activity.

**Figure 15 molecules-28-01690-f015:**
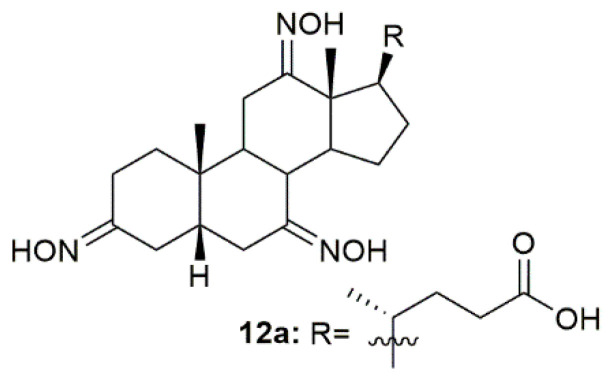
Bile acid derivative with anti-inflammatory activity.

**Figure 16 molecules-28-01690-f016:**
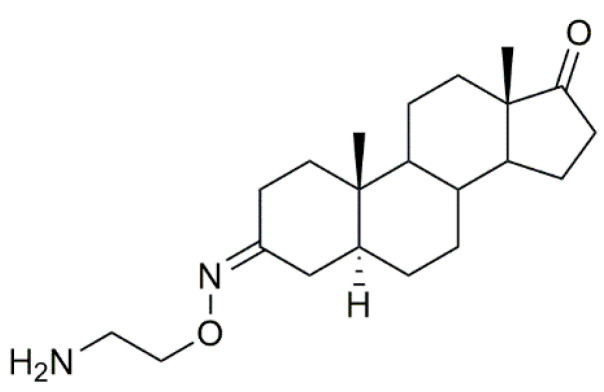
Istaroxime.

**Figure 17 molecules-28-01690-f017:**
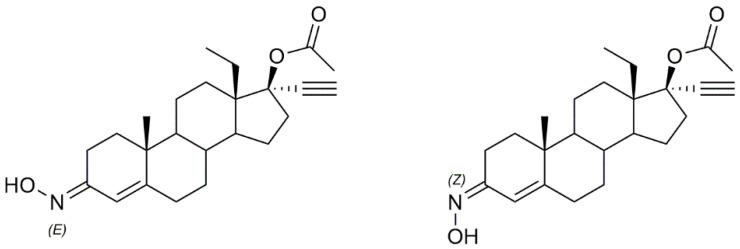
Norgestimate.

**Figure 18 molecules-28-01690-f018:**
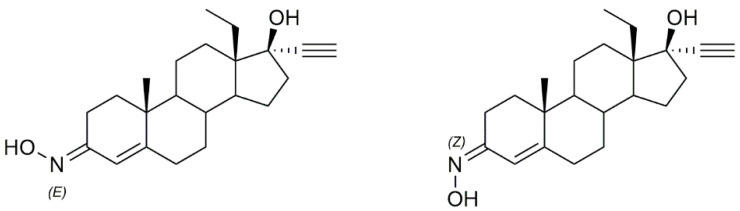
Norelgestromin.

**Figure 19 molecules-28-01690-f019:**
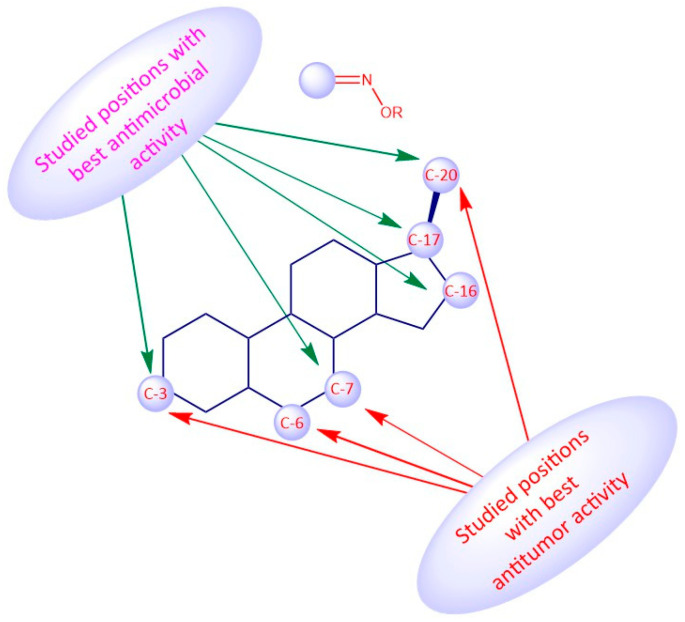
Overview of the most common SAR of steroidal oximes with antitumor and antimicrobial activity.

**Figure 20 molecules-28-01690-f020:**
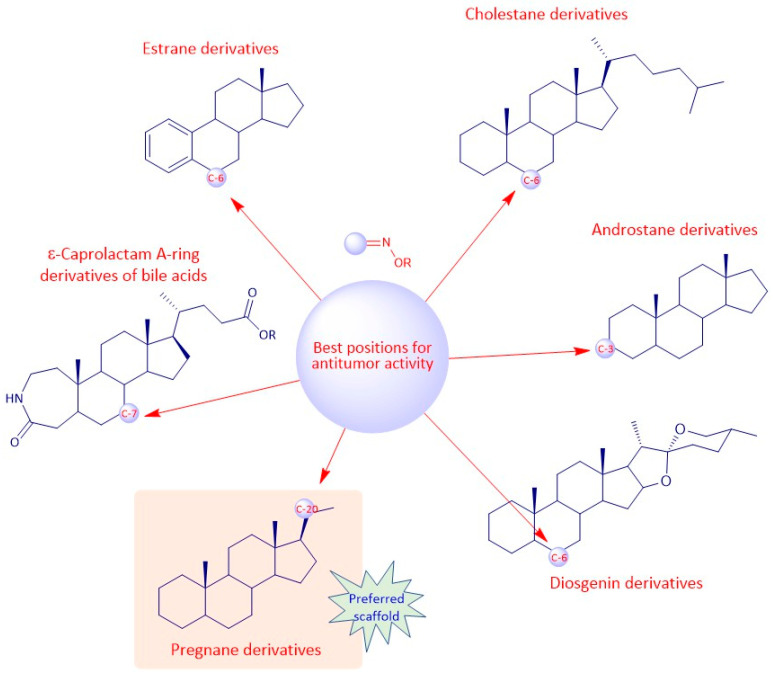
Most common SAR of steroidal oximes with antitumor activity.

**Figure 21 molecules-28-01690-f021:**
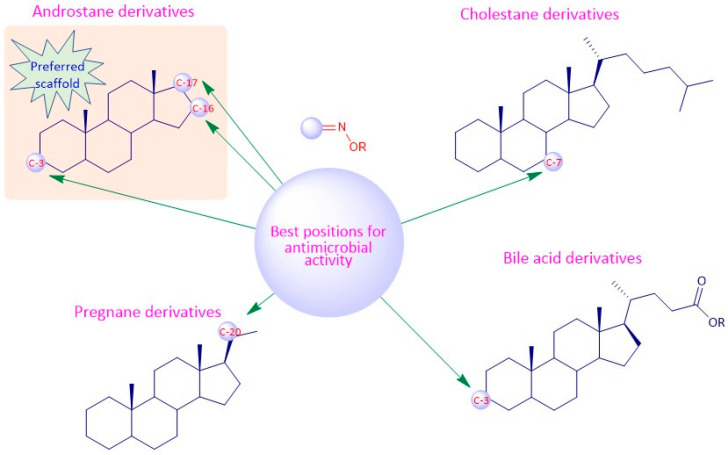
Most common SAR of steroidal oximes with antimicrobial activity.

**Table 3 molecules-28-01690-t003:** IC_50_ values (µM) of the synthesized pregnane oxime derivatives.

Cell Line	Compounds							
	3a	3b	3c	3d	3e	3f	3g	3h
HepG2	4.50	-	-	-	-	-	-	-
MDA-MB-231	6.76	-	-	-	-	-	-	-
T24	-	2.41	1.94	1.99	2.62	1.90	-	-
HT29	-	-	-	-	-	-	2.35	2.37
HCT15	-	-	-	-	-	-	0.31	0.65
SF295	-	-	-	-	-	-	1.67	5.44
HOP62	-	-	-	-	-	-	3.57	0.81
A549	-	-	-	-	-	-	3.56	7.17
MCF7	-	-	-	-	-	-	0.60	1.91
Ref.	[58]	[10]					[59]	

These values were obtained through different techniques such as MTT, MTS, and SRB assays.

**Table 4 molecules-28-01690-t004:** IC_50_ values (µM) of the synthesized cholestane oxime derivatives.

Cell Line	Compound															
	4a	4b	4c	4d	4e	4f	4g	4h	4p	4q	4r	4s	4t	4u	4v	4w
HeLa	35.2	20.7	-	-	-	22.8	5.6	-	-	-	-	-	-	-	-	-
K562	28.1	11.2	-	-	-	-	-	-	-	-	-	-	-	-	-	-
PBMC	110	34.6	>100	>100	>100	-	-	-	-	-	-	-	-	-	-	-
MCF7	-	-	8.2	9.5	7.9	-	-	-	-	-	-	-	-	-	-	-
MGC7901	-	-	-	-	-	12.8	16.3	-	-	-	-	-	-	-	-	-
SMMC7404	-	-	-	-	-	17.6	17.9	-	-	-	-	-	-	-	-	-
GNE2	-	-	-	-	-	12.1	-	-	-	-	-	-	-	-	-	-
SPC-A	-	-	-	-	-	74.5	-	-	-	-	-	-	-	-	-	-
Tu686	-	-	-	-	-	24.9	-	-	-	-	-	-	-	-	-	-
PC3	-	-	-	-	-	14.5	-	15.0	-	-	-	-	-	-	-	-
HT29	-	-	-	-	-	10.6	-	11.9	-	-	-	-	-	-	-	-
SH-SY5Y	-	-	-	-	-	-	-	16.8	-	-	-	-	-	-	-	-
HepG2	-	-	-	-	-	-	-	13.2	-	-	-	-	-	-	-	-
A549	-	-	-	-	-	-	-	15.0	11.6	1.06	11.5	11.5	11.7	19.4	11.5	2.22
ARPE19	-	-	-	-	-	-	-	23.7	-	-	-	-	-	-	-	-
BJ	-	-	-	-	-	-	-	17.1	-	-	-	-	-	-	-	-
HCT116	-	-	-	-	-	-	-	-	2.32	0.21	1.15	1.15	2.33	1.94	11.5	2.22
PSN1	-	-	-	-	-	-	-	-	2.32	1.06	11.5	23.1	>23.3	19.4	11.5	2.22
T98G	-	-	-	-	-	-	-	-	23.2	23.2	>23.1	23.1	23.3	19.4	11.5	2.22
Ref.	[46]		[47]			[48,49]		[51]	[54]							

**Table 5 molecules-28-01690-t005:** IC_50_ values (µg/mL) of the synthesized cholestane oxime derivatives.

Cell line	Compound																			
	4i	4k	4l	4m	4n	4o	4x	4z	4aa	4ab	4ac	4ad	4ae	4af	4ag	4ag	4ah	4ai	4aj	4ak
P388	1.25	1.25	1.25	0.25	0.25	0.5	-	-	-	-	-	-	-	-	-	-	-	-	-	
A549	1.25	1.25	1.25	0.13	0.13	0.13	-	-	-	-	-	-	-	-	-	-	-	-	-	-
HT29	1.25	1.25	1.25	0.25	0.25	0.25	-	-	-	-	-	-	-	-	-	-	-	-	-	-
MEL28	2.5	1.25	1.25	0.13	0.13	0.13	-	-	-	-	-	-	-	-	-	-	-	-	-	-
Sk-Hep-1	-	-	-	-	-	-	43	43	20.1	37	45	25	76.8	24	57	34.5	39.5	33.7	35.4	48.8
H292	-	-	-	-	-	-	59.5	59.5	26.2	37	62.5	46	70	33	76	53	41.9	34.2	65.8	78.9
PC3	-	-	-	-	-	-	44	44	32.5	40.5	41.5	76	>90	36	66	52	49.8	103	60.1	62.7
Hey-1B	-	-	-	-	-	-	37	49	26.3	45	53	38	78	37	51	45	47.9	45.7	56.3	61.5
Ref.	[67]	[68]					[70]	[71]									[72]			

**Table 6 molecules-28-01690-t006:** - IC_50_ values (µM) of the synthesized cholestane oxime derivatives.

Cell line	Compound											
	4al	4am	4an	4ao	4ap	4aq	4ar	4as	4at	4au	4av	4aw
HeLa	-	-	-	9.6	14.3	25.3	12.5	-	-	-	-	9.1
MGC7901	-	-	-	6.5	27.7	>100	33.3	-	-	-	-	-
SMMC7404	-	-	-	12.9	>100	62.2	10.2	-	-	-	-	-
GNE2	-	-	-	-	-	-	-	-	-	-	-	11.3
PC3	-	-	-	-	-	-	-	39.3	31.0	10.8	12.9	-
SGC-7901	13.2	32.3	26.2	-	-	-	-	-	-	-	-	-
Bel-7404	11.0	7.4	15.0	-	-	-	-	-	-	-	-	-
LNCaP	-	-	-	-	-	-	-	>100	26.2	44.8	13.9	-
Ref.	[73]			[74]				[75]				[60]

These values were obtained through different techniques such as MTT, KBR, alamar blue and SRB assays.

**Table 7 molecules-28-01690-t007:** IC_50_ values (µM) of the synthesized diosgenin oxime derivatives.

Cell Line	Compounds						
	5a	5b	5c	5d	5e	5f	5g
CaSki	10.51	48.18	-	-	-		
HeLa	10.9	41.61	37.5	-	-	11	19
MDA-MB-231	-	-	22.3	9.3	9.3		
HCT15	-	-	36.2	-	-		
MCF7	-	-	-	78.5	78		
A549						20	44
HBL-100						19	78
SW1573						22	>100
T47D						12	29
WiDr						14	17
Ref.	[78]		[79]	[80]		[81]	

These values were obtained through different techniques such as crystal violet staining and XTT assays.

**Table 8 molecules-28-01690-t008:** MIC and MBC (mg/mL) values of the synthesized androstane oxime derivatives. Adapted from [86].

Compound		MIC and MBC (mg/mL)							
		*S. a.*	*MRSA*	*L. m.*	*P. a.*	*P.aO1*	*E. c.*	*E. c. res*	*S. t.*
**7a**	MIC	-	0.020	0.15	0.10	0.10	0.060	0.15	0.20
	MBC	-	0.040	0.30	0.15	0.15	0.080	0.30	0.30
**7b**	MIC	0.20	0.030	0.15	0.08	0.10	0.04	0.08	0.15
	MBC	0.30	0.040	0.30	0.15	0.15	0.08	0.15	0.30
**7c**	MIC	-	0.20	0.30	0.15	-	0.15	-	0.30
	MBC	-	0.30	0.60	0.30	-	0.30	-	0.60
**7d**	MIC	0.30	0.075	0.30	0.10	0.10	0.20	-	0.30
	MBC	0.60	0.15	0.60	0.15	0.15	0.30	-	0.60
**7e**	MIC	0.10	0.0037	0.10	0.075	0.050	0.037	0.015	0.15
	MBC	0.15	0.015	0.15	0.15	0.075	0.075	0.037	0.30
Streptomycin	MIC	0.10	0.10	0.15	0.10	0.05	0.10	0.10	0.10
	MBC	0.20	-	0.30	0.20	0.1	0.05	0.20	0.20
Ampicillin	MIC	0.10	-	0.15	0.30	0.2	0.15	0.20	0.10
	MBC	0.15	-	0.30	0.50	-	0.20	-	0.20

*S. a.*—*S. aureus*; *MRSA*—methicillin resistant *S. aureus*; *L. m.*—*L. monocytogenes*; *P. a.*—*P. aeruginosa*; *P.aO1*—*P. aeruginosa* resistant; *E. c.*—*E.coli*; *E. c. res*—*E. coli* resistant; *S. t.*—*S. typhimurium*.

**Table 9 molecules-28-01690-t009:** MIC and MFC (mg/mL) values of the synthesized androstane oxime derivatives. Adapted from [86].

Compound		MIC and MFC (mg/mL)							
		*A. fum.*	*A. v.*	*A. o.*	*A. n.*	*T. v.*	*P. f.*	*P. o.*	*P. v.c.*
**7a**	MIC	0.037	0.015	0.015	0.037	0.007	0.015	0.02	0.05
	MFC	0.075	0.037	0.037	0.075	0.015	0.075	0.037	0.037
**7b**	MIC	0.015	0.015	0.007	0.037	0.007	0.015	0.037	0.05
	MFC	0.037	0.075	0.015	0.075	0.015	0.037	0.075	0.075
**7c**	MIC	0.075	0.015	0.075	0.15	0.10	0.15	0.20	0.20
	MFC	0.15	0.037	0.15	0.30	0.15	0.30	0.30	0.30
**7d**	MIC	0.015	0.037	0.10	0.075	0.015	0.15	0.075	0.15
	MFC	0.037	0.15	0.15	0.15	0.037	0.30	0.15	0.30
**7e**	MIC	0.037	0.037	0.075	0.075	0.050	0.075	0.15	0.10
	MFC	0.075	0.075	0.15	0.15	0.075	0.15	0.30	0.15
Ketoconazole	MIC	0.20	0.20	0.15	0.20	1.00	0.20	1.50	0.30
	MFC	0.50	0.50	0.20	0.50	1.50	0.50	1.50	0.30
Bifonazole	MIC	0.15	0.10	0.15	0.15	0.15	0.20	0.10	0.10
	MFC	0.20	0.20	0.20	0.20	0.20	0.25	0.25	0.20

*A. fum*.—*A. fumigatus*; *A. v.*—*A. versicolor*; *A. o.*—*A. ochraceus*; *A. n.*—*A. niger*; *T. v.*—*T. viride*; *P. f.*—*P. funiculosum*; *P. o.*—*P. ochrochloron*; *P. v.c.*—*P. verucosum* var. *cyclopium*.

**Table 10 molecules-28-01690-t010:** MIC values of (µg/mL) of the synthesized pregnane oxime derivatives. Adapted with permission from Ref [88], 2023, Ana S. Pires.

Compound	MIC (µg/mL)					
	*B. subtilis*	*S. epidermidis*	*P. vulgaris*	*P. aeruginosa*	*A. niger*	*P. chrysogenum*
**8b**	≤64	≤128	>512	≤256	≤128	≤128
**8c**	≤128	≤64	>512	≤128	≤64	≤256
**8d**	≤64	≤64	>512	≤256	≤128	≤128
**8e**	≤128	≤128	>512	≤128	≤64	≤256
**8f**	≤64	≤128	>512	≤256	≤128	≤128
**8g**	≤128	≤64	>512	≤512	≤128	≤256
**8h**	≤64	≤128	>512	≤256	-	≤256
**8i**	≤128	≤256	>512	≤128	≤256	-
**8j**	≤128	≤64	>512	≤256	≤64	≤128
**8k**	≤512	≤128	>512	≤512	≤128	≤256

Bacteria: *Bacillus subtilis* (MTCC 619), *Staphylococcus epidermidis* (MTCC 435), *Proteus vulgaris* (MTCC 426), *Pseudomonas aeruginosa* (MTCC 424). Fungi: *A. niger* (MTCC 1344), *P. chrysogenum* (MTCC 947).

**Table 11 molecules-28-01690-t011:** MIC values (µg/mL) of cholestane oxime derivatives.

Microorganism	Compound				
	9d	9e	9f	9g	9h
*C. albicans*	64	8	32	32	
*C. neoformans* (CN1)	4	4	4	2	
*B. poitrasii*	32	8	16	16	
*Y. lipolytica*	8	64	16	16	-
*F. oxysporum*	>64	16	8	8	-
HSV-1 strain B2006	-	-	-	-	19.5
HSV-2 strain G	-	-	-	-	17.9
PrV strain RC79	-	-	-	-	17.2
HSV-1 strain F	-	-	-	-	16.7
Ref.	[91]				[92]

## Data Availability

Not applicable.

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
