# Peer review of "The Structural Diversity and Biological Activity of Steroid Oximes"

_molecules, 2023, doi:10.3390/molecules28041690_

Round 1

Reviewer 1 Report

In general, the review focuses on the activity of oxime compounds from several reported studies without discussing the molecular mechanisms involved in such activities or the role of the functional group of interest (molecular interactions) of the synthesized compounds.  The activities mentioned are shared by many other non-oxime compounds. Accordingly, the manuscript shows only the screening results for such compounds which may have limited benefit to the reader in understanding the importance of oximes. It would also be useful if the manuscript discusses the common SAR of the compounds for each of the mentioned activities as well as the effect of other properties as Log P, half-life or number of ionizable groups, etc. on the activity. This can guide researchers in future work to design more potent or selective compounds.

There are other minor comments in the manuscript that needs to be addressed as well:

1-      In the abstract, it is mentioned that oximes are formed by the reaction between an aldehyde and ketone with an amine. This should be corrected to aldehyde or ketone, since the reaction requires only one of these two compounds with the amine.

2-      “Moreover, this steroidal oxime was not toxic to mouse macrophages, which indicates that this compound might be selective towards cancer cells.”

Why did selectivity assessment focus on macrophages? normally, researchers test the compound on several cancer and normal cells to evaluate the selectivity of cytotoxic activity? Is this related to the compound mechanism of action. Please explain this in the manuscript.

3-      “Further evaluation revealed that compound 4b exerted its cytotoxicity by inducing apoptosis in HeLa cells. On the contrary, compound 4b showed no evidence of apoptosis or necrosis, which can be explained by the difference in the Z/E stereochemistry in the hydroximine group of both compounds.”

It seems that there is a mistake in the previous sentences as both are talking about the same compound 4b while mentioning different effects.

Reviewer 2 Report

This paper reviewed the synthesis and biological activities of steroidal oximes. Especially, the structure and different biological activities, including anticancer, anti-inflammatory, antibacterial, antifungal and antiviral, are highlighted. In addition, the mechanisms of action and agents in clinic were summarized. The topic fits the scope of this journal and may inspire more drug discovery research in the future. In general, the manuscript is well-organized and the references are supportive to the conclusions. The key issues are required to be addressed before its publication on Molecules.

1. The available target information of these steroidal oximes are required to be involved, which will benefit the other drug discovery programs better.

2. The structures of these steroidal oximes are required to be listed as a general scaffold with different substituents to reduce the repeating core structure.

3. The structure-activity relationship of steroidal oximes for different biological activities are suggested to be summarized in a figure illustration.

4. The different structural characteristics of steroidal oximes for different biological activities are suggested to be summarized in a figure illustration.
